# Leveraging Multi-Modal Saliency and Fusion for Gaze Target Detection

**Athul M. Mathew**                                                    AMATHEW@ELM.SA
**Arshad Ali Khan**                                                      ARKHAN@ELM.SA
**Thariq Khalid**                                                      TKADAVIL@ELM.SA
**Faroq AL-Tam**                                                        FALTAM@ELM.SA
**Riad Souissi**                                                       RSOUISSI@ELM.SA
*Elm Company, Saudi Arabia*

## Abstract

Gaze target detection (GTD) is the task of predicting where a person in an image is looking. This is a challenging task, as it requires the ability to understand the relationship between the person's head, body, and eyes, as well as the surrounding environment. In this paper, we propose a novel method for GTD that fuses multiple pieces of information extracted from an image. First, we project the 2D image into a 3D representation using monocular depth estimation. We then extract a depth-infused saliency module map, which highlights the most salient (*attention-grabbing*) regions in image for the subject in consideration. We also extract face and depth modalities from the image, and finally fuse all the extracted modalities to identify the gaze target. We quantitatively evaluated our method, including the ablation analysis on three publicly available datasets, namely VideoAttentionTarget, GazeFollow and GOO-Real, and showed that it outperforms other state-of-the-art methods. This suggests that our method is a promising new approach for GTD.

**Keywords:** gaze target detection, gaze-following, 3D gaze, free-viewing, saliency, depth map, 3D projection, point cloud, multi-modal, fusion

## 1. Introduction

It is general phenomenon that human gaze acts as a natural cue which provides rich contextual information on the attention of individuals when it comes to social interactions, engagements and communication. Gaze is a fundamental human communication mean, since it can express emotions, feelings, and intentions (Lund, 2007). Human beings have a remarkable capability to follow the gaze of others to understand their gaze target, understand whether a person is gazing at them and determine the attention of others (Chong et al., 2018). GTD, also known as *gaze-following*, is an active research area and can have a wide range of applications, including human-computer interaction, educational assessment, treatment of patients with cognitive or neurological disorders such as early diagnosis of ADHD (Attention Deficit Hyperactivity Disorder) in children and so on.

Gaze estimation (GE) utilizes eye or facial images of a person to estimate the direction of gaze for the person (Huang et al., 2017) (Krafka et al., 2016). These methods utilize facial features of the person, such as the eyes, nose, and mouth, to estimate the three-dimensional orientation (i.e. *Yaw, Pitch* and *Roll*) of the face and predict the gaze direction. These methods can be less accurate, especially in challenging conditions such as low lighting or when the person is wearing glasses. Typical gaze detection and tracking systems often

require a calibration step, where the user is asked to look at various points before usage of such a system. Also, such gaze tracking systems are constrained because they are designed to monitor the gaze of one person when the person is well situated within the confined space of monitoring for the gaze tracking system. An example of such a system is a driver attention or fatigue detection system in vehicles where the system is expected to monitor the gaze of the driver seated on the driving seat.

Unlike GE, the GTD of a given person in an image involves learning relationship between the relative position of the person within the scene and the surrounding objects that lie within the field-of-view of that person. A robust and scalable gaze target assessment system is needed to identify the salient objects that people are likely looking at. While significant progress has been made in GTD from images, incorporating depth-related contextual cues remains a challenge. Such cues can enhance the accuracy and robustness of GTD, but reconstructing these cues from 2D images in multiple scenarios is difficult. This problem is further exacerbated when detecting the gaze target when the person is looking at a point on a smartphone screen (Zhang et al., 2015), or when predicting fixation on an object when the person is looking at a salient object within or outside the frame (Chong et al., 2020). In this paper, we focus on in-frame GTD due to time and scope constraints.

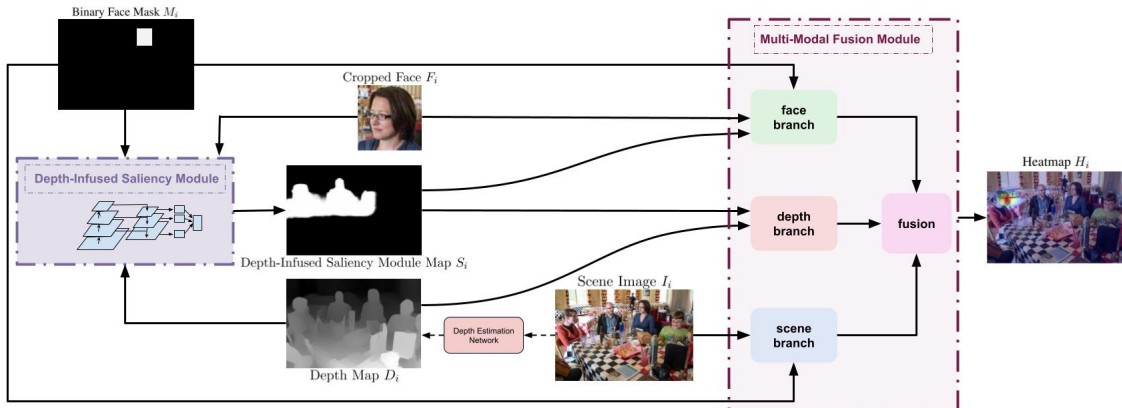

Figure 1: Overview of our multi-modal saliency and fusion architecture for gaze target detection.

Our GTD architecture consists of two modules, as shown in Figure 1. Firstly, **D**epth-**I**nfused **S**aliency **M**odule (DISM) extracts a map (binary mask) that highlights objects and artifacts that are in the line-of-sight of the subject's gaze within the scene. This map is predicted from the scene features that lie along the subject's field-of-view, based on contextual cues such as the subject's depth and spatial positioning in the scene. The second module, **M**ulti-**M**odal **F**usion (MMF), blends the DISM map with rich representations of scene, depth, and head features. This fusion process creates a unified representation of the scene that incorporates information from multiple sources. Together, the two modules work together to infer the gaze fixation of a given subject within the image. We further elaborate this method in Section 3. It is important to mention that many visual-attention-based saliency models (Itti and Koch, 2001) (Judd et al., 2009) focus on identifying visual gaze fixations of a person *free-viewing* a natural image. In GTD, people in the picture might fixate on objects even when they are not the most salient (Recasens* et al., 2015). Our work

goes beyond this, since we learn gaze target with respect to a person's viewpoint, and the learnt saliency map adapts based on the person's location and facial orientation within the image. In this research, our network in DISM aims to identify salient scene artefacts specific to the person in consideration within the image and this may not necessarily be the *free-viewing* saliency. Additionally, our MMF module introduces dedicated attention layers to distill the learnt saliency across face, scene, and depth branches and fuses the multi-modal embeddings effectively. Such multimodal design leads to achieving better performance when compared to those presented in similar contemporary prior-art research.

The remainder of this paper is organized as follows: Section 2 reviews state-of-the-art GE and GTD methods. Section 3 lays out our methodology, including our proposed architecture for extraction of DISM map and MMF. Section 4 discusses extensive experimental results and evaluation against established benchmarks using Area Under the Curve (AUC), Distance and Angular metrics. In Section 5, we summarize our research findings and, Section 6 sets out our goals for extending this work in the future.

## 2. Related Work

The study of automatic gaze analysis can be divided into two categories: gaze estimation (GE) and gaze target detection (GTD) (Chong et al., 2020) (Fang et al., 2021) (Recasens et al., 2017). GE estimates the direction of a person's gaze, typically in 3D, and does not necessarily focus on precisely locating the object of their interest (Zhang et al., 2018)(Guo et al., 2023). Methods such as Parks et al. (2015) estimate the gaze direction and do not identify the objects that are being attended to. On the other hand, Liu et al. (2020) uses a head-mounted eye-tracker to estimate the user's point of gaze. Similarly Thakur et al. (2021) also elaborate a method to detect where each person in the scene is looking by fusing videos and Inertial Measurement Unit (IMU) data. However, both these methods focus on GTD from the first-person viewpoint. There have been significant developments in gaze and saliency mapping, but robust 3D gaze orientation determination is still a challenging problem. Chong et al. (2018) rely on a 3D gaze angle regression model for GTD. Subsequently, Chong et al. (2020) extends the work to include temporal information to directly output an estimate of gaze uncertainty. In this paper, we focus on GTD with in-the-wild images, captured from a third-person viewpoint.

GTD has been evolving given the adaptation of computer vision technologies in human gaze research. It has become evident that in domains where close-level iris/eye tracking is not possible, head pose is the most important feature for estimating human focus of attention, along with other semantic information. While modelling 3D gaze requires additional human annotations (Jin et al. (2022)), models trained on such datasets still struggle to generalize to common scenarios. Fang et al. (2021) have pinpointed three significant issues in previous research. Firstly, most research works explore the gaze direction in 2D representation without encoding the depth modality. Secondly, salient object searching from 2D visual cues without depth understanding. Finally, learning mapping functions directly from head position to gaze direction without considering the relationship between eyes and head. Thus a comprehensive understanding of 3D scenes is essential to identify candidate objects lying at different depth layers along the subject's gaze direction.

While many approaches learn the mapping function from head features to gaze direction using 2D visual cues (Recasens* et al., 2015) (Chong et al., 2018) (Chong et al., 2020) (Recasens et al., 2017), estimating depth information from a RGB image is essential to accurately predicting the gaze target. Fang et al. (2021) and Tonini et al. (2022) introduced a dedicated depth branch to embed depth-related cues within their architecture. However, conventional depth estimation is a challenging task given the ill-posed nature of estimating real depth from a single RGB image. In the context of GTD, for example, using monocular depth estimation there may be multiple solutions in terms of estimating the distance from a viewpoint for a given target. Recently, deep neural networks have mitigated this problem by exploiting multiple visual cues such as relative size, brightness, patterns, and vanishing points extracted from an RGB image (Shim et al., 2023). New deep-learning frameworks for head localization and pose estimation on depth images are being used to tackle issues arising from poor illumination conditions, occlusion, and dynamic scenes (e.g., in low light and with illumination changes during the day). Moreover, Shim et al. (2023) propose a transformer-based approach called Relative Depth Transformer (RED-T), which uses relative depth as guidance in self-attention, such that the model assigns high attention weights to pixels of close depth and low attention weights to pixels of distant depth. However, transformer-based models typically require longer training and inference time than CNN counterparts, especially when a multimodal solution is adopted in the inference pipeline. Another challenge is the 3D reconstruction of the scene once the depth estimation is done, in order to place objects in 3D space and align them with their estimated depths.

Saliency mapping, in depth estimation, is another aspect that involves identifying and highlighting the most visually significant regions in a scene contributing to the depth perception element. Researchers like Recasens* et al. (2015) and Chong et al. (2020) present methods that completely discard utilization of depth modality within their networks which may prevent the GTD algorithms from identifying salient regions in a scene from the perspective of a human observer. Tonini et al. (2022) include a dedicated branch for depth modality, where the depth is processed as pixels in 2D image space. Fang et al. (2021) extract a depth-based attention map, however, they utilize only the coarse depth features for extraction of the saliency map.

One of the challenges, arising from integrating 2D annotation with 3D scene models for GTD is the semantic understanding and interaction with the scene. Bao et al. (2022) propose a GTD method that explicitly models 3D scenes using only 2D gaze annotations. Their research is particularly interesting because it considers 3D geometry to model the scene for GTD. However, this method assumes that the front-most object is always the salient object and this may not be valid at all times.

Tu et al. (2022) propose Human Gaze Target detection Transformer (HGTTR), which simultaneously detects multiple human head locations and their associated gaze targets at once in an image (instead of salient object detection and gaze prediction separately). This approach is more computationally efficient than the traditional two-stage head location and gaze target detection pipeline. However, HGTTR reports high false positives in images with a single human gaze target, making it less attractive for users evaluating gaze target detection datasets such as GazeFollow.

In terms of interesting real-world use cases, Senarath et al. (2022) proposed Retail Gaze, and Tomas et al. (2021) proposed GOO which are datasets for GTD in real-world retail

environments. There has also been some work in the context of classroom gaze measurement use-case. Ahuja et al. (2021) develop a new computer vision system that powers a 3D "digital twin" of the classroom. GTD is performed by post-processing the estimated gaze orientation of the face and ArUco markers (Garrido-Jurado et al., 2014) placed on objects around the classroom. Furthermore, Ömer Sümer et al. (2021) experimented with multimodal engagement analysis from facial videos in the classroom.

## 3. Our Methodology

Figure 1 outlines our method which extracts the DISM map $S_i$ using depth map $D_i$, binary face mask $M_i$, and cropped face $F_i$. The scene image $I_i$ along with other modalities $D_i$, $F_i$, $M_i$, $S_i$ are fused to estimate the gaze target point of any given person in an image. Furthermore, our methodology has been clearly explained in Algorithm 1.

---

**Algorithm 1:** Method for Gaze Target Detection

---

**Input:** Image stream $I_1, \ldots, I_n$
**Output:** Heatmap $H_i$
**Given:** Depth-Infused Saliency network $f_{ds}$, Scene Branch $f_s$, Depth Branch $f_d$,
         Face Branch $f_f$, Fusion Branch $f_n$
**Function** Depth_Infused_Saliency($D_i,M_i,F_i$):
    |        **return** $S_i = f_{ds}(D_i, M_i, F_i)$
**End Function**
**Function** Multi_Modal_Fusion($S_i,I_i,D_i,M_i,F_i$):
    |        scene_features $= f_s(\text{concatenate}(I_i, M_i))$
    |        depth_features $= f_d(\text{concatenate}(D_i, S_i))$
    |        face_features $= f_f(F_i)$
    |        modulated_scene_features $=$ modulate_scene(scene_features, face_features, $S_i$)
    |        modulated_depth_features $=$ modulate_depth(depth_features, face_features, $M_i$)
    |        **gaze_fixation** $= f_n(\text{modulated\_scene\_features, modulated\_depth\_features})$
    |        return **gaze_fixation**
**End Function**
**Function** GazeTargetDetection($I_i$, $D_i$, $M_i$, $F_i$):
    |        $S_i =$ Depth_Infused_Saliency($D_i,M_i,F_i$)
    |        gaze_fixation $=$ Multi_Modal_Fusion($S_i,I_i,D_i,M_i,F_i$)
    |        return **gaze_fixation**
**End Function**

**for** $i \leftarrow 1$ **to** $n$ **do**
    |        Extract $D_i$, $M_i$, $F_i$ from $I_i$
    |        $H_i =$ GazeTargetDetection($I_i$, $D_i$, $M_i$, $F_i$)
**end**

---

### 3.1. Depth-infused Saliency Module (DISM)

The depth-infused saliency network, $f_{ds}$, uses a Feature Pyramid Network (FPN) architecture (Lin et al., 2017) to learn high-level semantic saliency for the subject of interest. The network takes a concatenated 7-channel input of the scene depth map $D_i$, binary head position mask $M_i$, and face image $F_i$. $M_i$ and $D_i$ encode the subject's relative three-dimensional position in the scene. $F_i$ helps the model to focus on scene artefacts that lie along a projection plane originating from the subject's facial position along the depth axis and are directed parallel to the facial orientation vector. The network finally predicts a DISM map, $S_i$ that highlights the most likely gaze fixation artefacts for the subject in the scene. An overview of the depth-infused saliency network $f_{ds}$ is shown in Figure 2.

It is our intention to simplify the learning objective of DISM and utilize it to provide rich cues for the MMF module. We bin the human gaze direction $\theta$ along the depth plane $\theta_d$ into *forward* $(\theta_{d_f})(90°)$, *intermediate-forward* $(\theta_{d_{if}})(45°)$, *same-plane* $(\theta_{d_s})(0°)$, *intermediate-backward* $(\theta_{d_{ib}})(-45°)$ and *backward* $(\theta_{d_b})(-90°)$ directions. The gaze direction along the image plane $\theta_{xy}$ is binned into *lower-right* $(\theta_{xy_{lr}})(30°)$, *straight* $(\theta_{xy_s})(90°)$, *lower-left* $(\theta_{xy_{ll}})(150°)$, *upper-left* $(\theta_{xy_{ul}})(220°)$ and *upper-right* $(\theta_{xy_{ur}})(320°)$ directions. At any instance, the 3D-gaze angle will comprise an image plane component and a depth plane component. That is, $\theta = [\theta_{xy}, \theta_d]$, where $\theta_{xy} \in \{\theta_{xy_{lr}}, \theta_{xy_s}, \theta_{xy_{ll}}, \theta_{xy_{ul}}, \theta_{xy_{ur}}\}$ and $\theta_d \in \{\theta_{d_f}, \theta_{d_{if}}, \theta_{d_s}, \theta_{d_{ib}}, \theta_{d_b}\}$.

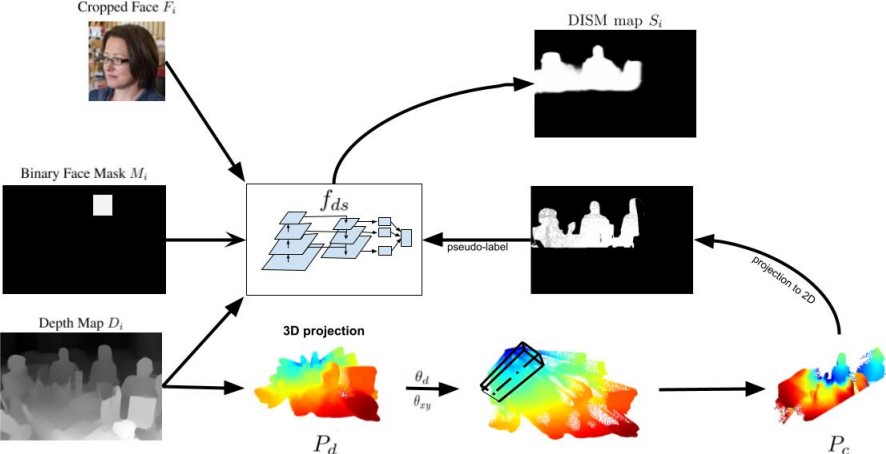

Figure 2: Overview of DISM. We take 3D projection of depth map $P_d$ alongside gaze binning parameters $\theta_d$ and $\theta_{xy}$ to extract a sub-collection of filtered 3D points $P_c$. The re-projection of $P_c$ back to the image-plane serves as pseudo-labels for the FPN network, $f_{ds}$. The network provides a representation of the learned DISM map $S_i$.

### 3.1.1. Data pre-processing

Pseudo-labels for DISM require relative depth between face and target point, computed using monocular depth estimation techniques such as Ranftl et al. (2020) and Ranftl et al. (2021). Depth plane gaze angle $\theta_d$ is extracted using average depth of face $d_f$ and target points $d_t$, and empirically setting depth plane binning thresholds $\gamma_1$ and $\gamma_2$ to 3 and 10 respectively.

$$\theta_d = \begin{cases} \theta_{d_s} & \text{, if } d_f - d_t < \gamma_1 \\ \theta_{d_{if}} & \text{, if } \gamma_1 < d_f - d_t < \gamma_2 \\ \theta_{d_{ib}} & \text{, if } \gamma_1 < d_t - d_f < \gamma_2 \\ \theta_{d_f} & \text{, if } d_f - d_t > \gamma_2 \\ \theta_{d_b} & \text{, if } d_t - d_f > \gamma_2 \end{cases} \tag{1}$$

The gaze direction along the image plane $\theta_{xy}$ is extracted from the pixel positions of the eye $(e_x, e_y)$ and the gaze target $(g_x, g_y)$. Given $\theta_{xy} \in \{\theta_{xy_{lr}}, \theta_{xy_s}, \theta_{xy_{ll}}, \theta_{xy_{ul}}, \theta_{xy_{ur}}\}$, the

image angle $\alpha$ is discretized which assumes one of the values within $\theta_{xy}$. $\alpha$ is computed as:

$$\alpha = \arctan \frac{g_y - e_y}{g_x - e_x} \tag{2}$$

*where* the fraction calculates the gradient between the eye location and gaze fixation points.

### 3.1.2. Depth-infused saliency map

The dataset $A$ comprises of $N$ images such that $A = \{I_i, D_i, M_i, F_i\}_{i=1}^N$, where $I_i \in \mathbb{R}^{H_i \times W_i \times 3}$ is $i$-th image in the dataset. $H_i$ and $W_i$ denote the width and height of image. $D_i$ is the depth map of $I_i$. The binary mask of the head position of the subject within the scene is denoted as $M_i$ and $F_i$ is the cropped face of the subject. The DISM map for all images is represented as $S = \{S_i\}_{i=1}^N = \{\{s_i^m\}_{m=1}^{H_i \times W_i}\}_{i=1}^N$ where $s_i^m \in \{0, 1\}$ denote presence of $m$-th pixel in DISM map of $i$-th image, with $m = [1, ..., H_i \times W_i]$. We can represent the network $f_{ds}$ which predicts the DISM map $S_i$ as :

$$S_i = f_{ds}(D_i, M_i, F_i) \tag{3}$$

In order to extract the ground truth DISM map $S_i$ (pseudo-labels), the depth map is projected onto a 3D grid representation using focal length $(f_x, f_y)$ and optical centre $(c_x, c_y)$ of the depth camera parameters from Places dataset. The extrinsic parameters are assumed to be an identity matrix. Let $P_d$ be the collection of 3D projection points of the depth map $D_i$. For every pixel location $(a, b)$ of the depth map, $p$ is a point within the collection $P_d$ such that :

$$p = \begin{cases} p_x = \frac{(j - c_x) D_i[a,b]}{f_x} \\ p_y = \frac{(i - c_y) D_i[a,b]}{f_y} \\ p_z = D_i[a, b] \end{cases} \tag{4}$$

A cuboid aligned along $\theta_{xy}$ in the XY plane and $\theta_d$ in the Z plane is projected from the face position in 3D space. The orientation of the 3D projection cuboid is determined using the image plane and depth plane gaze angle parameters $\theta_{xy}$ and $\theta_d$ obtained from Equation (1) and (2), respectively. $P_c$ is the collection of 3D points within the volume of the projected cuboid, where $P_c \subset P_d$. The collection, $P_c$ is then re-projected back to the image plane as a binary mask using the depth camera parameters to finally derive the DISM map $S_i$. The ground truth saliency mask, $S_i$, and prediction saliency mask, $\hat{S}_i$ are trained with the objective of minimizing the Jaccard distance (JD). We have opted for JD as it is considered suitable for binary segmentation tasks or mask comparison, especially in our case where precise delineation of regions matters a lot. The metric provides normalized measures of IOU along with computational efficiency and interpretability benefits which allows for meaningful comparison across different scales. The objective function to minimize JD is given by $L_j$ as :

$$L_j(S_i, \hat{S}_i) = 1 - \frac{(S_i \cdot \hat{S}_i) + \epsilon}{(S_i + \hat{S}_i - S_i \cdot \hat{S}_i) + \epsilon} \tag{5}$$

where $\epsilon$ prevents zero division.

Our method in DISM generates 3D point clouds and embeds complete 3D information when modelling the DISM map and makes no assumptions regarding the spatial relationship of the salient points with respect to the subject. Our network supports deep supervision and is trainable end-to-end. Furthermore, our network is groundbreaking in its approach, as it addresses the task of modelling the likelihood of a subject's gaze location within the scene as if it were a scene segmentation problem.

### 3.2. Multi-Modal Fusion (MMF) module

The MMF network, $f_{mm}$ is shown in Figure 3. It outputs a heatmap $H_i \in \mathbb{R}^{H_i \times W_i}$ reflecting the probability of the gaze fixation point for a subject within the scene. We utilize the scene image $I_i$, the depth map $D_i$, the cropped face $F_i$, the binary face position mask $M_i$, and the DISM map $S_i$ to identify the gaze target. We can represent the network $f_{mm}$ mathematically as:

$$H_i = f_{mm}(I_i, D_i, M_i, F_i, S_i) \tag{6}$$

where $S_i$ is obtained from the DISM network $f_{ds}$ in Equation (3).

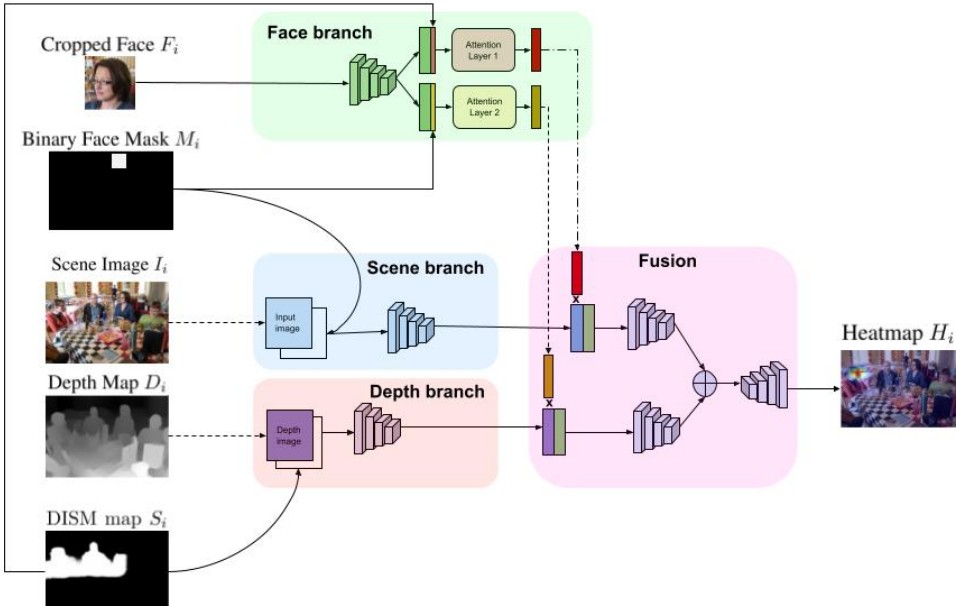

Figure 3: Our MMF module comprises three branches - face, scene, and depth. The three branches are fused in the Fusion module. The output of the module is a 2D Heatmap $H_i$ superimposed on the scene image $I_i$ here for visualization.

**Face Branch** $f_f$ extracts facial features of dimension $1024 \times 7 \times 7$ from the cropped face image $F_i$ using the face backbone. It then average-pools the extracted features to $e_i^F$, which has dimensions $1024 \times 1 \times 1$. $e_i^F$ is then separately processed using a set of linear projections to learn the attention weights. Attention Layer 1 embeds the depth relevance by concatenating $e_i^F$ with the max-pooled and flattened DISM map $e_i^S$, which represents the salient depth information in the scene. Attention Layer 2 embeds the spatial relevance of

the face by concatenating $e_i^F$ with the max-pooled and flattened binary face position mask $e_i^M$. Both attention layers ($attn_i^S$ and $attn_i^M$) are represented as a set of linear projections $f_S$ and $f_M$, respectively. These linear layers are then passed through a softmax function that applies weightage to spatial and depth-relevant cues within the image.

$$
\begin{aligned}
attn_i^S &= \Phi(f_S(e_i^F \oplus e_i^S)) \\
attn_i^M &= \Phi(f_M(e_i^F \oplus e_i^M))
\end{aligned}
\tag{7}
$$

where $\oplus$ denote concatenation operation and $\Phi$ denote softmax function.

**Scene Branch** $f_s$ branch takes as input the scene image $I_i$ and the binary face position mask $M_i$. The two inputs are concatenated and passed through the scene backbone to extract the scene embedding $e_i^I$, where each embedding has a dimension of $1024 \times 7 \times 7$. $e_i^I$ is then modulated by $attn_i^S$. The dimension of $attn_i^S$ is $1 \times 7 \times 7$. The modulated scene embedding $e_i^{I*}$ has a dimension of $1024 \times 7 \times 7$ and is given by :

$$
e_i^{I*} = e_i^I \otimes attn_i^S
\tag{8}
$$

**Depth Branch** $f_d$ takes the depth map $D_i$ and DISM map $S_i$ as inputs. The two inputs are concatenated and passed through the depth network backbone. The output depth embedding $e_i^D$ from the network has a dimension of $1024 \times 7 \times 7$. The depth embeddings are also modulated by $attn_i^M$ having a dimension of $1 \times 7 \times 7$. The modulated depth embedding $e_i^{D*}$ with the dimension of $1024 \times 7 \times 7$ is given by :

$$
e_i^{D*} = e_i^D \otimes attn_i^M
\tag{9}
$$

where $\otimes$ represents elementwise multiplication operation.

The modulated scene embeddings $e_i^{I*}$ and depth embeddings $e_i^{D*}$ are concatenated with face embeddings $e_i^F$ and are separately encoded using scene and depth encoders $f_e^I$ and $f_e^D$. The encodings are fused by summation and finally passed on to a decoder $f_d$ for predicting the gaze target heatmap $H_i$. The MMF network $f_{mm}$ can thus alternatively be represented as :

$$
H_i = f_{mm}(I_i, D_i, M_i, F_i, S_i) = f_d \left[ f_e^I(e_i^{I*} \oplus e_i^F) + f_e^D(e_i^{D*} \oplus e_i^F) \right]
\tag{10}
$$

The ground-truth gaze heatmap, $\hat{H}_i$ is attained by overlaying a Gaussian weight centred around the target gaze point. The objective of the network is to minimize the Heatmap Loss $L_h$ which is computed using Mean Squared Error (MSE) loss for cases when the gaze target is present inside the frame for $N$ instances within the dataset.

$$
L_h(H_i, \hat{H}_i) = \sum_{i=1}^{N} (H_i - \hat{H}_i)^2
\tag{11}
$$

### 3.3. Implementation Details

We have implemented the training and inferencing pipeline of our model using the PyTorch framework. All inputs are normalized and resized to $224 \times 224$ pixels. DISM uses a Resnet-101 (He et al., 2015) backbone pre-trained on ImageNet (Russakovsky et al., 2015). It has

a 5-stage encoder design with 256 and 128 convolution filters in the FPN feature pyramid and segmentation blocks, respectively. All backbones in the MMF module are pre-trained similar to Chong et al. (2018), Chong et al. (2020) and Fang et al. (2021). The scene and depth backbones were pre-trained on the Places dataset (Zhou et al., 2014), and the head backbone was pre-trained on the Eyediap dataset (Funes Mora et al., 2014). The face, scene, and depth feature extractors use Resnet-50 (He et al., 2015) backbones. The network outputs a $64 \times 64$ gaze heatmap. We use random crop, colour manipulation, random flip, and head bounding box jittering for data augmentation during training. We train the DISM and MMF module on GazeFollow (Recasens* et al., 2015) until convergence; then fine-tune on VideoAttentionTarget (Chong et al., 2020). We also train the network from scratch on the GOO-Real (Tomas et al., 2021) dataset. We use the Adam optimizer (Kingma and Ba, 2014) with a learning rate of 0.00025 and a batch size of 48.

## 4. Experiments

We quantitatively and qualitatively evaluated our full model on the VideoAttentionTarget, GazeFollow and GOO-Real datasets. We followed the standard training/testing splits of all datasets for a fair evaluation. We demonstrate that our method surpassed the performance of prior methods across most metrics in Section 4.3. Moreover, we perform an ablation study in Section 4.4 to validate the effectiveness of each module within our architecture.

### 4.1. Datasets

**VideoAttentionTarget** dataset comprises 164,541 frame-level head bounding boxes with 109,574 in-frame gaze targets and 54,967 out-of-frame gaze annotations. 10 shows were kept aside as test split, which comprises of 31,978 gaze annotations. **GazeFollow** dataset comprises of 122,143 images and about 160,000 annotations of people head bounding boxes and their corresponding gaze points. **Gaze On Objects (GOO)** dataset focuses on the retail environment where several grocery items are placed on shelves to imitate a real grocery store. GOO comprises 192,000 synthetic images (*GOO-Synth*) and 9552 real images (*GOO-Real*).

### 4.2. Evaluation Metrics

We use three evaluation metrics in line with previous works such as Chong et al. (2018, 2020); Fang et al. (2021); Recasens* et al. (2015); Lian et al. (2019); Tonini et al. (2022) to assess our model's performance. In the GazeFollow dataset, the ground truth gaze target location is estimated by taking the average of the annotations provided by 10 different human annotators for each image and subject. **Area Under Curve (AUC):** We compare the flattened output gaze heatmap to the flattened binarized ground truth heatmap and plot the ROC curve using True Positive Rate and False Positive Rate. The AUC score is the area under this curve, with a score of 1.0 denoting perfect agreement of the prediction with the ground truth. **L2 Distance(Dist.):** The Euclidean distance between the ground truth target location and prediction heat map maximum is measured after normalizing the image height and width to 1. For the GazeFollow dataset, we also calculate the minimum distance (Min. Dist.) between the predicted gaze point and the 10 ground truth gaze target points for each subject. **Angular error(Ang.):** This metric reports the angular difference

between the predicted gaze direction and the ground truth gaze vector between face location and gaze point. We report all results from our experiments in Table 1 and Table 2.

### 4.3. Multi-Modal Saliency and Fusion Model Evaluation

We compare our model to several state-of-the-art architectures (Chong et al., 2020) (Recasens* et al., 2015) (Lian et al., 2019) (Chong et al., 2018) (Fang et al., 2021) (Jin et al., 2022) (Bao et al., 2022) (Tonini et al., 2022) for in-frame gaze target detection. We observed that the overall performance of our model is better on VideoAttentionTarget and GOO-Real, which have higher-resolution images than GazeFollow. The improved resolution translates to better depth map representations and generation of more accurate DISM maps from the 3D projections. See Figure 4 for visualizations of depth maps, DISM maps, MMF heatmaps, and predicted gaze target points for example cases from different datasets.

| **Input Image** | **Depth Map** | **DISM Map** | **MMF Heatmap** | **Target Prediction** |
|---|---|---|---|---|

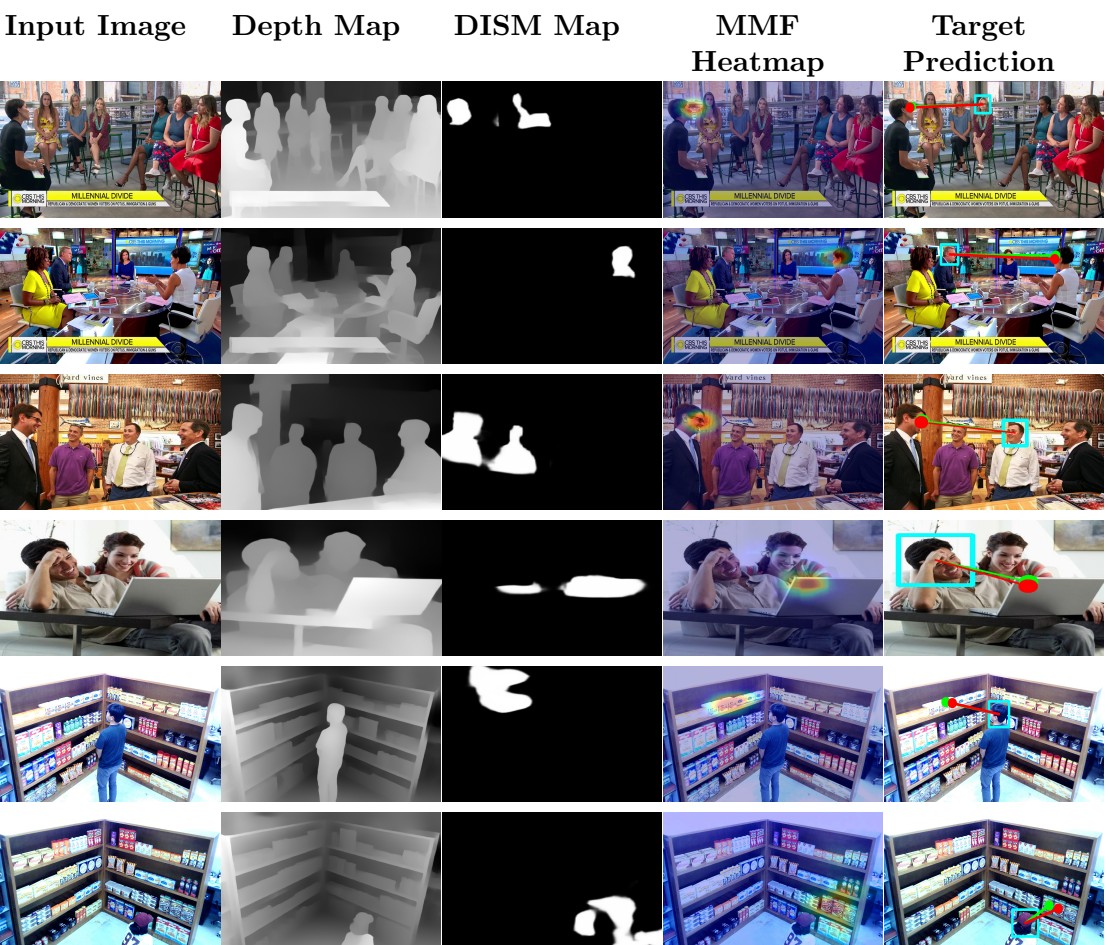

Figure 4: **Visualization results**. This figure shows examples from the VideoAttention-Target (first two rows), GazeFollow (middle two rows), and GOO-Real (last two rows) datasets. Each row shows the input image, depth map, DISM map, MMF heatmap, prediction result, and ground truth.

### 4.3.1. Evaluation on VideoAttentionTarget

Quantitative results on VideoAttentionTarget dataset are summarized in Table 1. *Random* denotes that the prediction is made with 50% chance by sampling the values randomly from a Gaussian distribution. In *Fixed bias*, the bias present in the dataset in terms of position of the faces and the relative gaze fixation points are taken into consideration. It is to be noted that the method in Chong et al. (2020) uses a spatio-temporal architecture for video-based prediction (denoted as VideoAttn). For fair comparisons, we also include the performance of its spatial-only counterpart (denoted as VideoAttn[†]).

**Ours**[‡] refer to the GazeFollow-trained model (no fine-tuning) and **Ours** refer to the GazeFollow-trained model that is fine-tuned on VideoAttentionTarget. It is interesting to note that the performance of our model surpasses the performance of Chong et al. (2020) without the inclusion of any temporal features for video-based gaze target detection. Our proposed architecture also exhibits impressive generalization capabilities. The model trained purely on GazeFollow (**Ours**[‡]) outperforms all other state-of-the-art architectures that were finetuned on VideoAttentionTarget in the *AUC* metric. Finally, the fine-tuning of our model on VideoAttentionTarget (**Ours**) results in new state-of-the-art scores for both *AUC* and *Dist.* metrics. Some qualitative images are shown in Figure 5.

Table 1: Evaluation on VideoAttentionTarget and Goo-Real. [*] indicates taken from Tomas et al. (2021). The best and second-best scores are highlighted in **teal** and red.

| Method | VideoAttentionTarget | | GOO-Real | |
|---|---|---|---|---|
| | AUC↑ | Dist.↓ | AUC↑ | Dist. ↓ |
| Random (Chong et al., 2020) | 0.505 | 0.458 | - | - |
| Fixed Bias (Chong et al., 2020) | 0.728 | 0.326 | - | - |
| Recansens et al. (Recasens[*] et al., 2015) | - | - | 0.850[*] | 0.220[*] |
| Lian et al. (Lian et al., 2019) | - | - | 0.840[*] | 0.321[*] |
| Chong et al. (Chong et al., 2018) | 0.830 | 0.193 | - | - |
| VideoAttn[†] (Chong et al., 2020) | 0.854 | 0.147 | - | - |
| VideoAttn (Chong et al., 2020) | 0.860 | 0.134 | 0.796[*] | 0.252[*] |
| Danyang et al. (Tu et al., 2022) | 0.893 | 0.137 | - | - |
| Fang et al. (Fang et al., 2021) | 0.905 | 0.108 | - | - |
| Jin et al. (Jin et al., 2022) | 0.901 | 0.116 | - | - |
| Bao et al. (Bao et al., 2022) | 0.885 | 0.120 | - | - |
| Tonini et al. (Tonini et al., 2022) | 0.940 | 0.129 | 0.918 | 0.164 |
| **Ours**[‡] | 0.958 | 0.123 | 0.876 | 0.208 |
| **Ours** | 0.964 | 0.100 | 0.954 | 0.130 |
| Human | 0.921 | 0.051 | - | - |

### 4.3.2. Evaluation on GOO-Real

Similar to the notation scheme above, **Ours**[‡] denote the performance of the model trained on GazeFollow and tested on GOO-Real. The performance of the model trained from scratch

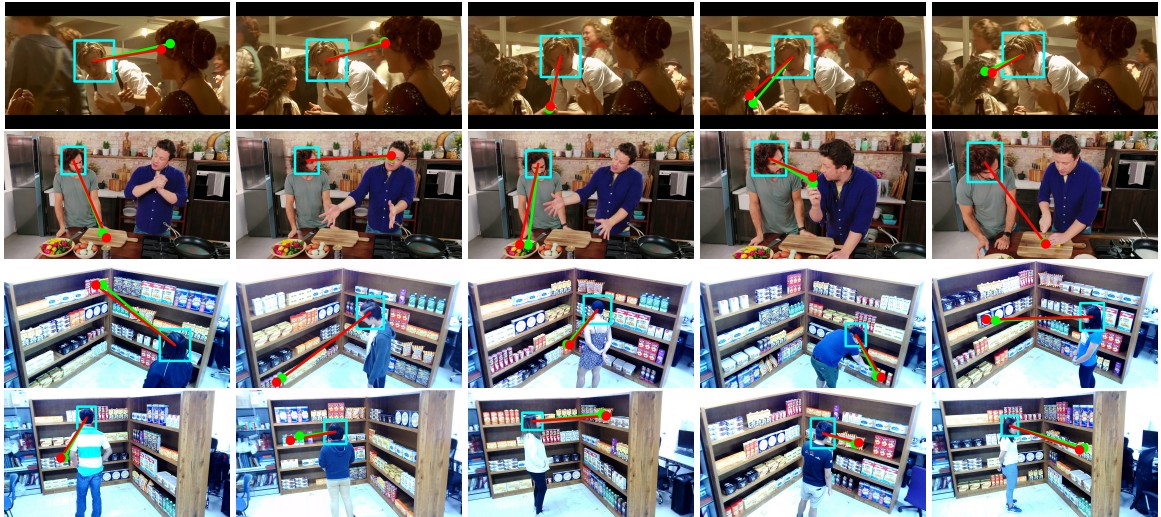

Figure 5: **Qualitative results**. The red and green lines denote ground truth and predictions respectively. The first two rows represent the changes in gaze target points of a subject from video sequences in VideoAttentionTarget. The last two rows are images from GOO-Real for varying head poses.

on GOO-Real is denoted as **Ours**. The authors of Tomas et al. (2021) trained and tested the architectures from Recasens* et al. (2015), Lian et al. (2019) and Chong et al. (2020) on GOO-Real dataset. The performance of these models is reported in Table 1. Our model trained on GazeFollow (**Ours‡**) was able to surpass the performance of Recasens* et al. (2015), Lian et al. (2019) and Chong et al. (2020) when tested on GOO-Real for both AUC and Dist. metrics even-though the model was not finetuned for retail gaze target detection using GOO-Real dataset. The model that was trained and tested on GOO-Real (**Ours**) sets the new benchmarks for both the AUC and Dist. metrics. Some qualitative images are shown in Figure 5.

### 4.3.3. EVALUATION ON GAZE FOLLOW

The quantitative results for the GazeFollow dataset are shown in Table 2. The entry labelled *Center* denotes the metrics are calculated by considering the gaze point to always be at the center of the image. Our model achieves new state-of-the-art in the AUC metric. Similar to Bao et al. (2022), we believe that $AUC$ is a better metric than *Dist.* and *Angle* for the GazeFollow dataset because the latter metrics are susceptible to errors introduced by averaging human annotations. For example, Figure 6 shows how averaging human annotations can lead to inconsistent estimates of the ground-truth gaze target point. Our example case highlights that averaging the human annotations causes the gaze point to drift away from the true gaze point and causes the averaged gaze point not to be consistently centred around the object that the person is looking at. We additionally include some challenging scenarios and failure cases from GazeFollow dataset in Figure 6. Our model generalises

Table 2: Evaluation on GazeFollow dataset

| Method | AUC↑ | Dist.↓ | Min. Dist.↓ | Angle↓ |
|---|---|---|---|---|
| Random | 0.504 | 0.484 | 0.391 | 69.0 |
| Center | 0.633 | 0.313 | 0.230 | 49.0 |
| Fixed bias | 0.674 | 0.306 | 0.219 | 48.0 |
| Recansens et al. (Recasens* et al., 2015) | 0.878 | 0.190 | 0.113 | 24.0 |
| Chong et al. (Chong et al., 2018) | 0.896 | 0.187 | 0.112 | - |
| Lian et al. (Lian et al., 2019) | 0.906 | 0.145 | 0.081 | 17.6 |
| Danyang et al. (Tu et al., 2022) | 0.917 | 0.133 | 0.069 | - |
| VideoAttn† (Chong et al., 2020) | 0.921 | 0.137 | 0.077 | - |
| Fang et al. (Fang et al., 2021) | 0.922 | 0.124 | 0.067 | 14.9 |
| Jin et al. (Jin et al., 2022) | 0.923 | **0.120** | **0.064** | 14.8 |
| Tonini et al. (Tonini et al., 2022) | 0.927 | 0.141 | - | - |
| Bao et al. (Bao et al., 2022) | 0.928 | 0.122 | - | **14.6** |
| **Ours** | **0.932** | 0.133 | 0.073 | 19.3 |
| Human | 0.924 | 0.096 | 0.040 | 11.0 |

well and identifies the correct target gaze heatmap bin consistently, thus achieving new state-of-the-art scores on the AUC metric.

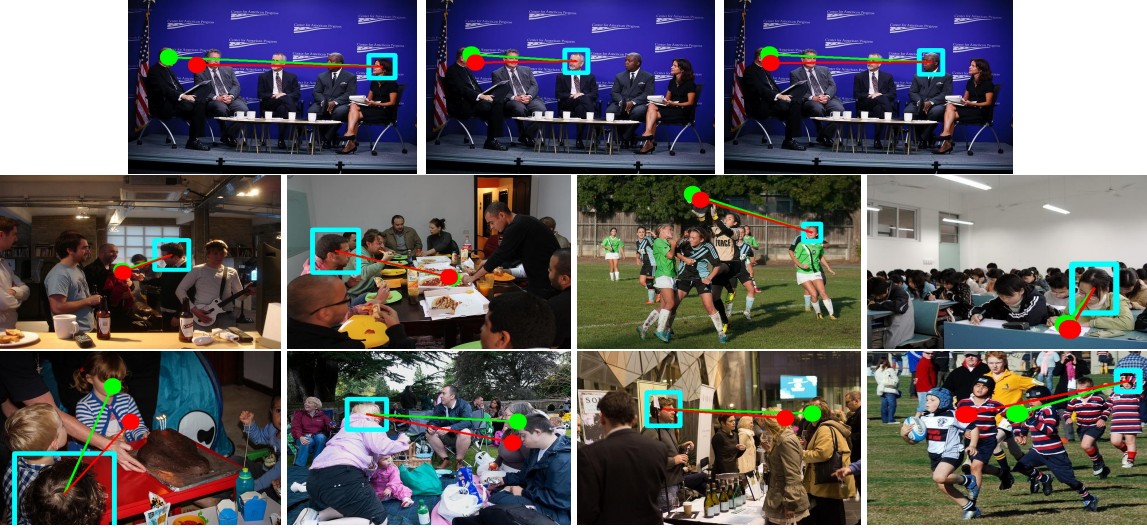

Figure 6: **Qualitative results**. Red, green and blue denote average human annotation, prediction, and head location, respectively. The first row shows inconsistent human annotation, while our model predicts the same gaze target for all three subjects. The second row shows our model's performance in challenging scenarios with complex backgrounds. The last row shows some failure cases, however, the predicted gaze targets contextually remain acceptable.

### 4.4. Ablation Study

In order to better understand the impact of various components within our system, we conducted further analysis using the GOO-Real dataset. Firstly, we removed the attention layers $attn_i^M$ and $attn_i^S$ from the network. This is reflected as injecting $e_i^D$ instead of $e_i^{D*}$ into $f_e^D$ and $e_i^I$ instead of $e_i^{I*}$ into $f_e^I$ (Attention-None). Next, we removed the DISM map, $S_i$, and replaced it with a uniformly weighted mask (DISM-None). We then removed all inputs in relation to the depth map $D_i$. We removed the Depth branch and $attn_i^S$ which modulates the scene embedding $e_i^I$ based on DISM map which is inherently learned from the depth map (Depth - None). Finally, we removed the Scene branch. This means that only the output of the depth encoder $f_e^D$ is fed to the decoder $f_d$ (Scene-None). We also evaluated the multi-modal fusion module using ground-truth DISM maps (DISM pseudo-labels). The results, sorted in the order of their performance are reported in Table 3. We notice that DISM contributes significantly to the overall model performance. We also show that all components of our network are necessary to attain exceptional performance.

Table 3: Ablation study on GOO-Real dataset

| Method | AUC↑ | Dist.↓ |
|---|---|---|
| DISM - None | 0.911 | 0.188 |
| Depth - None | 0.915 | 0.186 |
| Scene - None | 0.941 | 0.140 |
| Attention - None | 0.948 | 0.135 |
| Ours - All (DISM pseudo-labels) | 0.959 | 0.128 |
| **Ours - All** | **0.954** | **0.130** |

## 5. Conclusion

In this research, we have presented a GTD architecture comprising two key modules: Depth-Infused Saliency and Multi-Modal Fusion. The former focuses on identifying salient artefacts relevant to the subject within the scene image to generate the DISM map and the latter leverages the generated DISM map while fusing multiple modalities to generate the gaze target heatmap. This approach has proven to outperform similar contemporary research in terms of various state-of-the-art metrics. We presented challenging scenarios and failure cases to our model to test its generalization capabilities and it consistently pinpointed the gaze target correctly. This research represents a significant step forward in gaze target detection, offering a robust and effective approach to understanding human gaze within complex scenes.

## 6. Future Work

One of our next goals is to improve our network with a dedicated component for out-of-frame gaze detection. Furthermore, we envision extending our work into the field of education, aiming to enhance student-teacher interaction and engagement. Such systems necessitate a robust face detector. This inspires us to explore transformer-based networks and self-attention mechanisms, to concurrently identify faces and their associated gaze points.

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
