# OpenReview forum: "Leveraging Multi-Modal Saliency and Fusion for Gaze Target Detection"
_NeurIPS.cc/2023/Workshop/Gaze_Meets_ML — Gaze Meets ML 2023 Poster_

### Official Review · Reviewer_m6Ma · 2023-10-17
**Interesting method with thorough evaluation but lacking analysis and clarity**

**Rating:** 6
**Confidence:** 3

**Review:**

This work proposes a new method for gaze target detection (GTD). Core contribution seems to be the depth-infused saliency map, which seems to help improve the state-of-the-art performance of GTD on multiple datasets.

-The related work section seem to cover most relevant related work, however clear limitations that will be addressed in the given work are not properly discussed/summarized.

- In my opinion section 3.1 is not clear and should be iterated on, especially 3.1.1 and 3.1.2 need a clearer explanation on how exactly the pseudo-labels for the DISM map are calculated.
- I also wondered why they optimize DISM using the Jaccad distance, a brief discussion on the choice of loss function would be important (or ref to prev. work)
- At the end of 3.1 is again some literature review, which discussed limitations of previous work and motivates DISM. This should be incorporated into the previous work section (see my earlier point), i.e. first point out limitations and from there you can motivate your method and explain it.

- Would be nice if Figure 3 had a higher resolution. I also suggest to remove the DISM network from this figure and simply use the DISM map as an input to the MMF module. The overall architecture was already shown in Figure 1 and the DISM was explained. I see no need to show all the inputs to the DISM again here, it just increases visual clutter.

- The evaluation seems to be thorough, the authors compare on three commonly used datasets against many baselines from previous work.
- The analysis of the proposed method is a bit lacking. For example, the authors claim that the worse performance on the GOO dataset stems from the fact that the images have a lower resolution compared to the other papers, which is why the depth estimates are worse. It would be nice to validate this claim, for example by using a reduced resolution for the other datasets to see if performance drops accordingly.
- Evaluation performance of the DISM model itself would be interesting, how well can it predict the pseudo-labels?
- How would the GTD performance improve with ideal DISM maps, i.e. using the pseudo-labels as input?
- Given that the DISM seems to be the core contribution, its explanation, evaluation and analysis is lacking

Overall I would give the paper a borderline rating, slightly leaning towards accept. With the proposed changes and additional analyses mentioned I would rate the paper higher.

---

### Official Review · Reviewer_pHL1 · 2023-10-19
**a new network architecture for gaze target detection**

**Rating:** 8
**Confidence:** 4

**Review:**

The paper presents a new architecture for gaze target detection. It proposes two key modules: depth-infused saliency module and multi-module fusion module. The effectiveness of the proposed architecture is demonstrated by comparing with other state-of-the-art methods on several public datasets as well as doing ablation studies to show the impact of each module. The paper is written well and provides detailed information on the sections of literature review, method, experiments, and evaluation. It would be great if the authors can share their source code with the field to help promote reproducibility.

---

### Official Review · Reviewer_m7dR · 2023-10-24
**Well written contribution**

**Rating:** 8
**Confidence:** 4

**Review:**

The authors tackle the task of gaze target detection using a multi-staged architecture utilizing different information present in an image. The primary novelty of the approach is the introduction of a depth-infused saliency map module, targeted towards identifying viable candidates for the target’s gaze. The authors then propose fusing the results from various streams, termed as modalities, to perform fusion and make a heatmap prediction for the target gaze.
The overall paper is clearly written and well structured, with a strong introduction and related work section covering challenges of models utilizing 3D information in 2D modalities.
While the authors have clearly articulated the relationships between the various modules, the intuition behind how we can identify roughly accurate regions of interest for a subject without an initial gaze direction to get the DISM map can be further expanded upon.
The authors enumerate multiple steps required to train the DISM module effectively, however, there haven’t been any ablation studies to understand the impact of these various choices on the module’s performance or the overall performance of the model.
The authors current proposal requires various models pre-trained on other datasets e.g. Places, Eyediap. It opens up the debate of whether the improvements is just a function of architecture improvements, or inclusion of additional data and supervision from other sources.
Exploring more end-to-end approaches requiring less careful linking of such modules would be an interesting direction for the authors to explore in future works.
Overall, a well written paper with potential novelty in the form of the overall architecture.

---

### Meta-Review · Area_Chair_zDZS · 2023-10-26

**Recommendation:** Accept (Poster)
**Confidence:** 4

**Metareview:**

The paper introduces a novel approach using depth-infused saliency and multi-modal fusion modules for gaze target detection. The reviewers appreciated this work of introduction of depth-infused saliency. However, the authors need to address a few concerns raised by the reviewers, which the authors can address in the camera-ready version.

---

### Decision · Program_Chairs · 2023-10-26

Accept (Poster)